# Remote sensing measurements during PaCE 2022 campaign

Simo Tukiainen[1], Tuomas Siipola[1], Niko Leskinen[1], and Ewan O'Connor[1]

[1]Finnish Meteorological Institute, Atmospheric Composition Research

**Correspondence:** Simo Tukiainen (simo.tukiainen@fmi.fi)

**Abstract.** Continuous ground-based remote sensing measurements were conducted during the Pallas Cloud Experiment (PaCE) 2022 campaign. Remote sensing instruments, including two ceilometers (models CL31 and CL61, Vaisala Oyj), a Doppler cloud radar (model RPG-FMCW-94, RPG Radiometer Physics GmbH), and a Doppler wind lidar (model StreamLine XR, HALO Photonics), were deployed at Kenttärova, Finland, a measurement station that is part of the Pallas Atmosphere–Ecosystem Supersite. The instruments operated continuously throughout the entire campaign, with the exception of a few technical interruptions and brief maintenance periods. The PaCE 2022 remote sensing measurements provided vertical profiles of atmospheric targets with high temporal and vertical resolution, extending from the ground up to an altitude of 10–15 km depending on instrument. By combining the data from these instruments and a numerical weather prediction model, cloud micro- and macrophysical properties—such as ice water content, ice effective radius, and target classification—were retrieved using the Cloudnet methodology. The processed remote sensing data set complements the PaCE in situ measurements, providing valuable validation opportunities. The data set is available on the Cloudnet data portal at https://doi.org/10.60656/b3460d9d88d14fe6 (O'Connor and Hyvärinen, 2024).

## 1 Introduction

Measurements of cloud and aerosol micro- and macrophysical properties are crucial for understanding the complex cloud–aerosol interactions in the atmosphere. Both aerosols and clouds significantly influence Earth's climate system, and their incomplete representation leads to large uncertainties in climate and weather prediction models (e.g. Dagan et al., 2023; Shen et al., 2024). To address these knowledge gaps, comprehensive measurement campaigns, employing a broad suite of observational instruments, can provide new insights into the chemical and physical processes involved.

Arctic and Subarctic regions are particularly attractive locations for an intensive field campaign. High-latitude regions are projected to experience some of the most severe impacts of global warming (e.g. Neumann et al., 2019; Bailey et al., 2021), and these sensitive regions are often associated with the greatest modelling uncertainties. For example, low-level mixed-phase clouds, common in subarctic conditions, are notably difficult to model accurately (Schmale et al., 2021).

Since 2004, the Finnish Meteorological Institute (FMI) has organized a series of measurement campaigns known as the Pallas Cloud Experiment (PaCE), to study cloud–aerosol interactions and cloud microphysics (e.g., Doulgeris et al., 2022). The PaCE campaigns have been carried out in Pallas, northern Finland, in a cold and clean subarctic environment (Komppula

et al., 2005). The 2022 campaign took place from 12 September to 15 December 2022 and was the ninth PaCE since its inception.

Although the previous PaCE campaigns have focused mainly on studying cloud properties through in situ measurements (e.g., Kivekäs et al., 2009; Anttila et al., 2012; Doulgeris et al., 2023), some ground-based cloud remote sensing measurements were also performed at Kenttärova during the PaCE 2015, 2017, and 2019 campaigns. Remote sensing observations provide useful complementary data to in situ measurements. Using a combination of remote sensing instruments, cloud micro- and macrophysical properties can be retrieved through the entire vertical tropospheric profile using the so-called Cloudnet methodology (Illingworth et al., 2007; Griesche et al., 2024).

## 2   Site description

Kenttärova is a measurement station in northern Finland (67°59'N, 24°14'E; 347 m a.s.l.) situated in a forested environment below the tree line, near the Sammaltunturi fell (Fig. 1a). Kenttärova is part of the Pallas Atmosphere–Ecosystem Supersite hosted by FMI, and it lies approximately 5.5 km to the east and about 220 m lower than the main Pallas station located on the top of Sammaltunturi. Cloud remote measurements conducted at Kenttärova complement the in situ observations conducted at the summit of the fell. The top of the fell is occasionally inside a cloud, offering possibilities to study the same cloud via both measurement principles. A more comprehensive description of Kenttärova, as well as the general weather and cloud conditions in the Pallas region, can be found in Hatakka et al. (2003) and Lohila et al. (2015).

The Kenttärova station was established in 2002. The site features a 20 m high measurement tower for studying atmosphere–biosphere interactions above a spruce forest and is an ecosystem station for the Integrated Carbon Observation System (ICOS) network. Kenttärova also has a platform for ground-based remote sensing instruments (Fig. 1b) and a small cabin to house the power and telecommunication systems and servers needed to operate the instruments and transfer data (Fig. 1c). Continuous cloud remote sensing measurements at the site, with Cloudnet compliant remote sensing instrumentation, began on 23 August 2022.

Kenttärova will be one of the permanent cloud remote sensing sites in the Aerosol, Cloud and Trace Gases Research Infrastructure (ACTRIS,  Laj et al., 2024), which is currently in its implementation phase. As part of ACTRIS, cloud remote sensing measurements at Kenttärova will continue on a long-term basis in the future. The formal process to integrate Kenttärova into ACTRIS is underway and is expected to be completed within the next few years.

## 3   Instrumentation

During the PaCE 2022 campaign, the Kenttärova cloud remote sensing instrumentation included two ceilometers, a Doppler cloud radar, and a Doppler wind lidar (Table 1). Ceilometers are vertically pointing instruments. In the absence of a scanning unit, the cloud radar also pointed vertically. The Doppler lidar performed both vertical stare and azimuthal scan measurements at regular intervals.

As with all cloud remote sensing sites that comply with the ACTRIS requirements, Kenttärova's remote sensing instruments normally operate continuously and autonomously. Human intervention is usually required only when technical issues occur, such as system failures or prolonged power cuts. At regular intervals, measurement data are automatically transferred to the ACTRIS Cloudnet data portal (CLU, 2024c) for processing and archiving. Depending on the instrument, the data submission interval ranges between 15 minutes and 1 hour.

The Cloudnet data portal, hosted and developed by FMI, is physically located in Helsinki, Finland. After receiving the raw measurement data, the Cloudnet portal processes and archives it, and provides free and open access to raw data, processed products, and quicklooks. Data can be downloaded via both a graphical user interface and a REST API. The Cloudnet portal is designed to work in real time and typically processes and makes the processed products available within a few minutes after receiving the raw data. In addition, quality control is performed automatically during processing using the cloudnetpy-qc software (CLU, 2024d).

## 3.1 Ceilometers

During the PaCE 2022 campaign, FMI operated two ceilometers in Kenttärova: a Vaisala CL61 (O'Connor, 2024b) and Vaisala CL31 (O'Connor, 2024c). Both CL61 and CL31 are manufactured by the Finnish company Vaisala. The CL61 model measures both the profile of attenuated backscatter coefficient and depolarization ratio up to a range of 15 km, while the less-powerful CL31 model provides the profile of attenuated backscatter coefficient up to a range of about 7.6 km and has no polarimetric capabilities. Both instruments operate in the near infrared spectrum close to 910 nm wavelength. Although the ceilometer signal fully attenuates in thick liquid cloud layers, making it ideal for determining cloud base height, it can also provide profile information on aerosols, ice clouds and mixed-phase clouds in drier atmospheric conditions.

Both ceilometers were running on factory settings, without additional calibration factors applied during data processing. This can explain possible differences in the absolute value of attenuated backscatter coefficient between the two instruments. Having a properly calibrated ceilometer is important for deriving quantitative aerosol properties from the observed backscatter signal. However, because the Cloudnet processing scheme uses the ceilometer mainly to detect aerosol presence and liquid layers, small biases in the attenuated backscatter coefficient do not significantly impact the Cloudnet retrievals.

In the PaCE 2022 data set, CL61 was the primary instrument used for the synergetic geophysical products when data from both ceilometers were available. It is the newer model and has a higher range, higher sensitivity and better time and altitude sampling resolution compared to CL31 (Table 1). The time series of the noise-screened attenuated backscatter from CL31 and CL61 during PaCE 2022 are shown in Fig. 2c and Fig. 2d, respectively.

## 3.2 Doppler cloud radar

The FMI Doppler cloud radar (O'Connor, 2024a) deployed in Kenttärova during the PaCE 2022 campaign is an RPG-FMCW-94 model manufactured by the German company RPG Radiometer Physics GmbH. The radar operates in the W band at 94 GHz frequency (a wavelength of 3.2 mm) making it sensitive to relatively large atmospheric targets such as ice particles, cloud droplets, and insects. The FMI cloud radar supports dual polarization and operates with linear depolarization ratio (LDR)

mode. The measured LDR is a useful proxy for detecting the melting layer and insects from the data. The PaCE 2022 noise-screened cloud radar reflectivity is presented in Fig. 2a.

Additionally, the FMI cloud radar contains a single-channel passive radiometer at 89 GHz for measuring the liquid water path (LWP). Time series of the LWP from the PaCE 2022 campaign is shown in Fig. 2b. Standard Cloudnet instrumentation requires a dedicated multi-channel microwave radiometer (MWR) on site, but in certain atmospheric conditions a single-channel MWR is able to provide LWP with sufficient accuracy. Figure 5 shows a comparison of the LWP from a similar RPG-FMCW-94 cloud radar (Moisseev, 2024a) and a multi-channel RPG-HATPRO-G5 microwave radiometer (Moisseev, 2024b) in Hyytiälä, Finland, around 680 km south of Kenttärova, at the same time as the PaCE 2022 campaign. Measured LWP values over $0.1 \text{ kg m}^{-2}$ show a positive correlation ($r = 0.98$) with a mean difference of $7 \text{ g m}^{-2}$. For smaller values, the single-channel MWR measured a higher LWP with a mean difference of $21 \text{ g m}^{-2}$.

At 94 GHz, radar reflectivity measurements can be significantly influenced by attenuation due to atmospheric gases, liquid droplets, rain, and the melting layer. While the PaCE 2022 radar reflectivity data were corrected for gas and liquid attenuation, corrections for rain and melting layer attenuation were not applied. These attenuations can in principle be derived with the addition of a disdrometer, an in situ instrument that provides accurate data on the velocity and size distribution of precipitating particles. Unfortunately, no disdrometer was available during the campaign. As a result, certain properties, such as ice water content, could not be estimated accurately enough during periods of heavy rainfall.

It should be noted that the cloud radar was not specifically calibrated for the PaCE 2022 campaign using established methods such as those described by Toledo et al. (2020) or Jorquera et al. (2023). Furthermore, no disdrometer was available on site at the time to monitor or verify the calibration. Nevertheless, since the instrument had been recently acquired and a liquid nitrogen calibration was performed a month before the campaign, it is reasonable to assume that the radar reflectivity did not exhibit any significant bias during the observation period. One remaining potential source of error is radar mispointing. Since the cloud radar used in this study lacked a scanning unit, its vertical alignment relied solely on its internal tilt sensor. Although mispointing is generally less problematic for zenith-pointing cloud radars than for satellite-based or scanning weather radars, its impact should still be quantified. Identifying and correcting radar mispointing in the PaCE 2022 dataset is planned for future work.

### 3.3 Doppler wind lidar

The FMI Doppler wind lidar (O'Connor, 2024d) during the PaCE 2022 was a StreamLine XR model manufactured by HALO Photonics. The HALO Doppler lidar operates at 1565 nm and provides backscatter coefficient and Doppler velocity profiles up to a range of 12 km. Similarly to the CL61 ceilometer, the Doppler lidar can also measure the depolarization ratio. During the campaign, the lidar alternated between zenith-pointing stare measurements and velocity–azimuth display (VAD) scans at an elevation angle of 70 degrees. The VAD scans were used to derive horizontal winds as a function of altitude (Päschke et al., 2015; Pichugina et al., 2017). Additionally, the instrument performed a background noise measurement once per hour, which were used to correct the Doppler lidar backscatter coefficient measurements. The telescope focus function for the instrument

was determined using the methodology described in Pentikäinen et al. (2020), which enables the retrieval of the attenuated backscatter coefficient profile.

The HALO Doppler lidar measurements from the PaCE 2022 campaign were processed with Doppy (Leskinen, 2024), software for processing data from Doppler lidars written in Python and Rust, and based on the MATLAB code by Manninen et al. (2016) and the methodology of Vakkari et al. (2019). The Doppler velocity measured during the campaign is shown in Fig. 2e. The Doppler lidar products were generated as complementary data for the PaCE 2022 campaign and were not used in the processing of the synergetic geophysical products. Although the Doppler lidar attenuated backscatter coefficient could be

used in synergetic products, the two ceilometers offered higher temporal resolution due to their non-scanning operation.

### 3.4 Model data

The Cloudnet processing requires atmospheric profiles of temperature, pressure, and humidity to accurately derive higher-level geophysical products. These profiles are normally obtained from a numerical weather prediction (NWP) model or reanalysis data, but in principle measurements from a profiling instrument such as radiosonde or multi-channel microwave radiometer

can also be used.

The PaCE 2022 data set was processed using the Integrated Forecast System (IFS) model provided by the European Centre for Medium-Range Weather Forecasting (ECMWF). The IFS is an operational global numerical weather prediction system which provides hourly values for 137 height levels up to an altitude of 80 km. The current model horizontal resolution is about 9 km and the vertical resolution is better than 100 m in the lowest few km of the atmosphere. The model profiles were extracted

from the original global field at the closest grid point of the station. A full description of the IFS and the model updates over time can be found in the IFS documentation (ECMWF, 2024).

### 4 Derived products

To study cloud microphysical processes, remote sensing measurements must first be converted into meaningful geophysical properties. By using the synergy from co-located instruments that apply complementary measurement techniques, the full

atmospheric profile can be reliably characterized. The main geophysical variables derived from remote sensing measurements during the PaCE 2022 campaign are listed in Table 2.

The ceilometer and cloud radar measurements, together with the NWP model data, were combined using the so-called Cloudnet processing scheme (Illingworth et al., 2007). The main software for performing Cloudnet processing is the CloudnetPy Python package (Tukiainen et al., 2020). As a first step, the measurements are interpolated to a common time and height

resolution. Then, each data point, or pixel, is categorized for the presence of liquid droplets, falling hydrometeors, freezing temperature, melting ice particles, aerosols, and insects (Hogan and O'Connor, 2004). A single pixel may contain several categories at once.

Based on the categorization, pixels with drizzle, liquid, and ice are located. For drizzle pixels, parameters such as drizzle flux and mean particle size are retrieved from lidar and radar measurements (O'Connor et al., 2005). By combining radar reflectivity

and model temperature, the ice water content (Hogan et al., 2006) and ice effective radius (Griesche et al., 2020) are calculated for ice pixels. Likewise, radar reflectivity is used to calculate the droplet effective radius (Frisch et al., 2002) for liquid pixels. The liquid water content is derived by constraining the theoretical distributions of the adiabatic liquid water content with the observed LWP over the depth of a cloud containing liquid pixels (Illingworth et al., 2007).

A target classification product is also derived from the categorization. In the target classification product, each pixel is
160 assigned to one of 11 classes: "clear sky", "aerosols & insects", "insects", "aerosols", "melting & droplets", "melting ice", "ice & droplets", "ice", "drizzle & droplets", "drizzle or rain", or "droplets". An example of target classification from the PaCE 2022 campaign is shown in Fig. 3a. During the 24-hour period, liquid clouds and precipitation were detected below 2 km while ice clouds were seen above 4 km. Outside of precipitation, insects and aerosols were detected near the ground. Between 00:00 and 05:00 UTC, the ceilometer signal was fully attenuated very close to the ground due to low-level cloud or fog, but the radar
observations showed precipitation, and this was classified as "drizzle or rain".

The overall distribution of the target classification during the campaign is shown in Fig. 4. Typically, aerosols and liquid were only detected below 1 km while ice was detected higher up. Most aerosols and liquid were detected at the beginning of the campaign when surface temperatures were above freezing. Toward the end of the campaign, surface temperatures dropped below freezing and ice clouds were commonly detected also below 1 km.

One of the aims of Cloudnet is the evaluation of NWP models on properties such as cloud fraction, and liquid and ice water contents (Illingworth et al., 2007). Figure 3b shows the cloud fraction calculated based on classification and downsampled to the time and height resolution of the ECMWF IFS model. Compared with the corresponding cloud fraction from the IFS model (Fig. 3c), both are capable of detecting similar liquid and ice clouds, but, for example, the low-level cloud between 00:00 and 05:00 UTC is classified differently in the model and the Cloudnet classification.

A current limitation of the Cloudnet classification algorithm is that, while the cloud radar can reliably detect multiple cloud layers, the ceilometer signal will fully attenuate in the first encountered thick liquid cloud layer, rain, drizzle, or fog. To overcome this limitation, a machine-learning based method called VOODOO (Schimmel et al., 2022) has been developed. The VOODOO method uses the cloud radar Doppler spectra to estimate the probability of supercooled liquid in mixed-phase clouds. This probability can then be used in the categorization of liquid pixels. Figure 6 shows a comparison of the Cloudnet
classification using the standard method and the VOODOO method. The standard method detects less supercooled liquid than VOODOO and fails to identify any liquid above 3 km due to lidar attenuation. Under optically thick cloud conditions like this, VOODOO improves the standard classification, but more validation work is needed before it can be used operationally.

## 5 Data availability

The PaCE 2022 remote sensing data set can be found at https://doi.org/10.60656/b3460d9d88d14fe6 (O'Connor and Hyväri-
185 nen, 2024). The data set is published on the ACTRIS Cloudnet data portal (CLU, 2024a) under the Creative Commons Attribution 4.0 licence.

The data set includes noise-screened and harmonized instrument products, model products, and higher-level derived synergetic geophysical products. The data set contains 1,355 files and the total size of the set is 23.1 GB. The data set is organized into daily product files and stored in a NetCDF file format following the Cloudnet convention (CLU, 2024b).

Some products in the data set, such as Doppler wind lidar products and the VOODOO classification, are still under development, and their methods and implementations may be improved in the future. To distinguish them from the more mature products, they are labelled experimental. The raw measurements are not part of the data set but can still be accessed through the Cloudnet portal.

## 6   Code availability

The software for processing and visualizing the PaCE 2022 remote sensing data is written in Python and published on GitHub (CLU, 2024a). All Cloudnet software is developed as open source and released under the MIT licence. The ACTRIS Cloudnet data portal is also open source and follows the FAIR principles (Wilkinson et al., 2016).

## 7   Conclusions

Cloud remote sensing measurements were conducted during the PaCE 2022 measurement campaign. Two ceilometers, a Doppler cloud radar, and a Doppler wind lidar were deployed at the Kenttärova station, a measurement site part of the Pallas Atmosphere–Ecosystem Supersite. Kenttärova will be a permanent cloud remote sensing facility within the ACTRIS research infrastructure.

The remote sensing instruments operated continuously for the duration of the PaCE 2022 campaign, excluding a few technical breaks, providing vertical profiles of cloud properties with high vertical and temporal resolution up to 15 km. From the raw measurement data, geophysical synergetic products were derived using the Cloudnet methodology. The raw measurements and processed data products were made available through the Cloudnet data portal according to the FAIR principles.

The retrieved vertical profiles of cloud properties could be validated in the future using in situ measurements taken within the same cloud at the top of Sammaltunturi. This verification would be valuable, as the ACTRIS Cloudnet measurements are expected to serve as key validation data for satellite observations and various NWP models.

*Author contributions.*  ST wrote the manuscript together with TS. ST is the lead author of CloudnetPy and has been one of the main developers of the ACTRIS Cloudnet data portal since 2017. TS has developed the data portal and processing software since 2022. EO installed and operated remote sensing instruments at the site and is one of the original developers of the Cloudnet concept and methodology. NL generated the Doppler lidar products.

*Competing interests.*  There are no competing interests present.

*Acknowledgements.* The work was supported by the European Union Framework Programme for Research and Innovation, Horizon 2020 (ACTRIS IMP, grant no. 871115). The authors thank the Academy of Finland for supporting ACTRIS activities in Finland. The authors acknowledge ECMWF for providing the IFS model data.

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

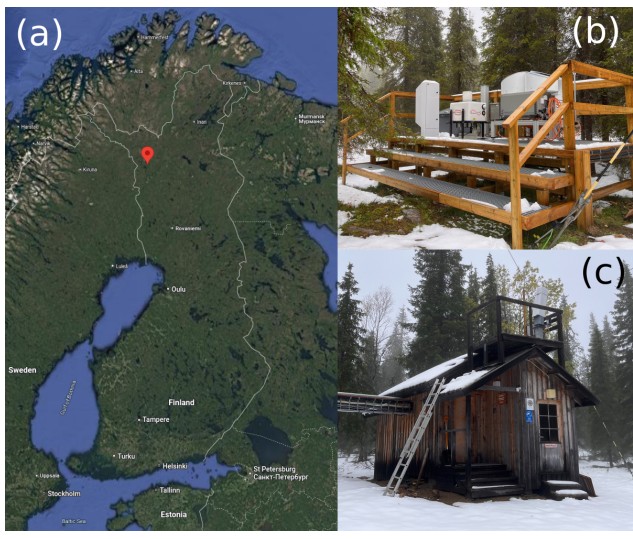

**Figure 1. (a)** Map of Finland showing the location of the Kenttärova station (© Google Maps 2024), **(b)** photograph of the platform with instruments, and **(c)** photograph of the cabin with a Vaisala CL31 ceilometer on the roof.

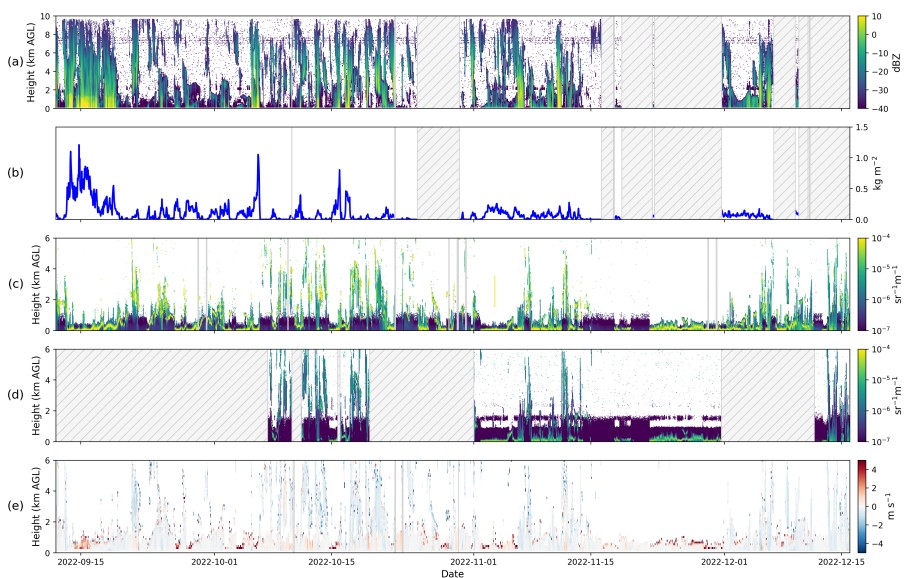

**Figure 2.** Example of noise-screened and hourly-averaged measurement data during the campaign: **(a)** RPG-FMCW-94 radar reflectivity, **(b)** RPG-FMCW-94 liquid water path, **(c)** CL31 attenuated backscatter coefficient, **(d)** CL61 attenuated backscatter coefficient, and **(e)** HALO Doppler velocity. The hatched regions indicate data gaps longer than 1 day.

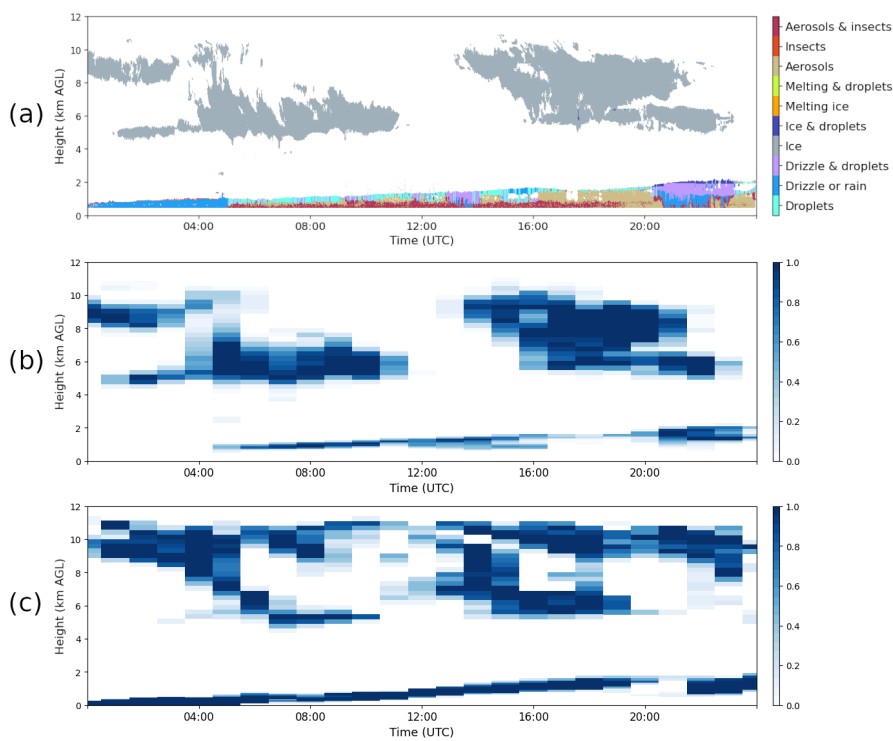

**Figure 3.** Example products on 23 September 2022: **(a)** Cloudnet target classification, **(b)** cloud fraction based on the classification, down-sampled to ECMWF IFS model level and time resolution, and **(c)** cloud fraction from the ECMWF IFS model.

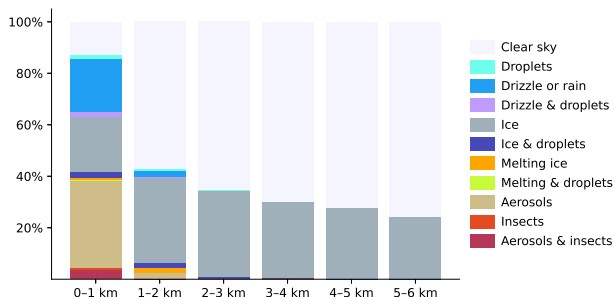

**Figure 4.** Distribution of target classes at different heights above ground level during the PaCE 2022 campaign.

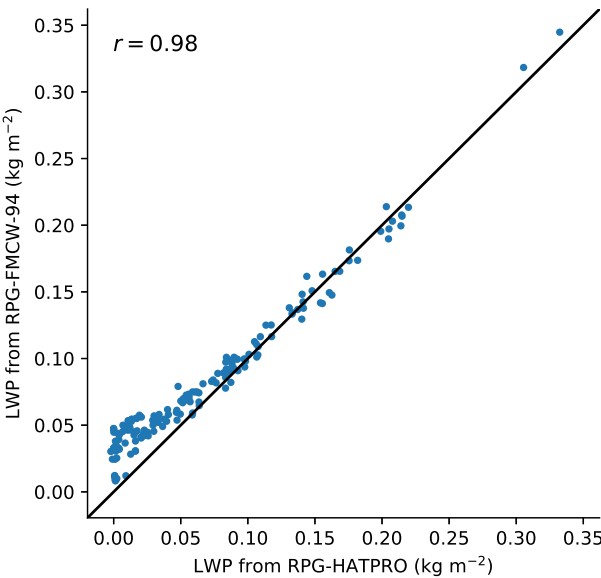

**Figure 5.** Comparison of liquid water path (LWP) hourly average from a single-channel microwave radiometer of a RPG-FMCW-94 cloud radar and a multi-channel RPG-HATPRO-G5 microwave radiometer in Hyytiälä at the same time as the PaCE 2022 campaign. Data was only available for the end the campaign from 2 to 15 December.

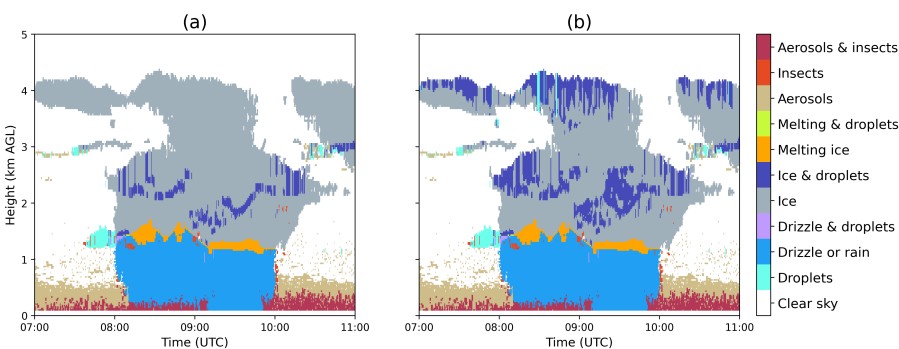

**Figure 6.** Example of the Cloudnet target classification on 24 September 2022 using **(a)** standard Cloudnet method and **(b)** VOODOO method.

**Table 1.** Cloud remote sensing instrumentation at Kenttärova during the PaCE 2022 campaign.

| Instrument | Primary measurements | Wavelength | Range | Range resolution | Time resolution |
|---|---|---|---|---|---|
| Vaisala CL31 ceilometer | Attenuated backscatter coefficient | 910 nm | 7.6 km | 9.8 m | 15 s |
| Vaisala CL61 ceilometer | Attenuated backscatter coefficient, linear depolarization ratio | 910 nm | 15 km | 4.8 m | 5 s |
| HALO StreamLine Doppler lidar | Attenuated backscatter coefficient, Doppler velocity, linear depolarization ratio | 1565 nm | 12 km | 120 m | 12 s |
| RPG-FMCW-94 cloud radar | Reflectivity factor, Doppler velocity, linear depolarization ratio, liquid water path | 3.2 mm | 12 km | 26–38 m | 4 s |

**Table 2.** Main geophysical variables derived from cloud remote sensing measurements with units, mean uncertainty during the campaign and retrieval method.

| Variable | Units | Uncertainty | Method |
|----------|-------|-------------|--------|
| Drizzle median diameter | m | 42 % | O'Connor et al. (2005) |
| Drizzle number concentration | $m^{-3}$ | 70 % | O'Connor et al. (2005) |
| Droplet effective radius | m | 31 % | Frisch et al. (2002) |
| Ice effective radius | m | 57 % | Griesche et al. (2020) |
| Ice water content | $kg\,m^{-3}$ | 83 % | Hogan et al. (2006) |
| Liquid water content | $kg\,m^{-3}$ | 11 % | Illingworth et al. (2007) |