# Peer review of "Remote sensing measurements during PaCE 2022 campaign"

_Earth System Science Data, 2024_

## Referee Comment (RC2)

Review of "Remote-sensing measurements during PaCE 2022 campaign" by Simo Tukiainen et al.

The present study discussed the important measurements over Kenttärova during the 9[th] PaCE campaign using cloud remote sensing instrumentation including two ceilometers, a Doppler cloud radar, and a Doppler wind lidar. The manuscript is well written discussing the campaign datasets, and methodology to estimate the cloud micro- and macrophysical properties briefly. The retrieved cloud products (ice water content, ice, and droplet effective radius) are beneficial to complement PaCE in-situ measurements. I believe this valuable dataset would also be helpful for the ongoing calibration and validation work for EarthCARE satellite data. In this context, the present study assumes its importance and thus I recommend the manuscript for publication. However, the authors missed some of the important details which I would suggest the authors implement in the revised manuscript.

Specific Comments:

1. The importance of the site: The author mentioned the importance of an intensive field campaign over the Arctic and Subarctic region in the introduction, but failed to connect it to their campaign site: Kenttärova. I understand the site is situated in a Subarctic region, but the site description lacks the importance of the site from the cloud remote sensing perspective. The author needs to mention why the location is strategic to establish a cloud remote sensing campaign over there. It can be done by a brief discussion of the cloud climatology over the site and the necessity of studying those clouds and their interaction with aerosols using cloud remote sensing data. If there is already a detailed description in other studies please mention the reference.

2. Quality control of any data set is very crucial. The cloudnetpy-qc software that the author mentioned, is a Python package implementing the Cloudnet processing scheme based on the study of Illingworth 2007. The author mentioned Doppler wind lider performed a background noise measurement once per hour, which was used to correct the measurements during data post-processing. Apart from this information, nothing is provided in the present study or the Documentation on the Cloudnet site. For example; do the authors use wind lidar or LDR from radar to screen out the insects? Therefore, I suggest the authors to be more elaborative about the criteria and methodology they have used for the quality control of each instrument. The authors can also add it as documentation in the Cloudnet Data Portal and mention the reference in the present study.

3. Further, the cloud classification methodology is adopted from Hogan and O'Connor (2006). Hogan and O'Connor were unable to distinguish supercooled drizzle from ice, but the present study shows the category of 'ice and droplets'. Also, the target classes shown in the present study are different than Hogan and O'Connor's (2006). A systematic explanation is necessary on how the authors are achieving those 11 classes.

If the quality control and cloud classification are two general methods used in the Cloudnet site for the stations with Cloud Remote sensing facility, then adding the details as a document in the site would be useful for the readers of the present study as well as the Cloudnet users.

4. All the derived products depend on the radar reflectivity. Hence, the radar reflectivity offset calibration is essential. Since the 94 GHz radar is highly attenuated to rain, and hail, and the authors have no option to check the offset due to the lack of any Disdrometer observations, they need to keep that in mind while validating with in-situ observations, and also the data need to be 'flagged' during heavy rain condition. Further, for vertically looking radar, mis-pointing could be an issue and it can lead to high bias to Doppler Velocity. I suggest the author discuss how they took care of the mis-pointing calibration of the radar.

5. In regard to estimating droplet effective radius, Frisch et al. 2000 study is limited to stratus cloud only. Does the present study also focus on the stratus clouds while estimating droplet effective radius? If so, how the authors are classifying different types of clouds?

6. The goal of comparing cloud fractions with the ECMWF IFS model is not clear. The model having very low resolution clearly could not capture all the cloud features and hence nowhere it have any similarity with the observation in Figure 3. In spite, I would suggest the authors keep the estimated ice and liquid water content in the middle and the effective radius at the bottom panel in that figure for the same example.

Minor Comments:
1. Please mention if all the heights discussed in the present study are above mean sea level or ground level.

2. In lines 84-85, I understand by 'large atmospheric targets' the authors meant compared to wind lider, but W-band radar would be attenuated for light rain, so please modify the line as 'making it sensitive to relatively large atmospheric targets such as ice particles, cloud droplets, and insects.'

4. In Figure 2. please mention in the caption, the time axis over which the parameters are averaged. Since the authors are showing data for 3 months, the radar reflectivity and also other parameters might be hourly or half-hourly averaged, so please mention that.

5. Modify the Figure 4 to include the y-tick label 100 %. Does the exclusion of 'clear sky' cause the first column below 100%?

6. What does it signify by 'Melting & droplets'? Does it quantify the ice that is melting near the bright band? There is no occurrence of 'Melting & droplets' in Figure 4, is it because the radar is not attenuation corrected for the melting layer and hence the use of the melting layer to detect melting ice and droplets is not possible in this study?

---

## Author Comment (AC1)

**Minor comments**

1. **Comment:**

   Overall, I think, the presentation of the data set could benefit from validation and comparison to other measurements, e.g., collected during previous campaigns. This could help the user, for example, to assess the liquid water path retrieved by the cloud radar, as the standard Cloudnet instrumentation for this quantity is a microwave radiometer.

   **Response:**

   Remote sensing data collected during the previous PaCE campaigns were quite limited, and as already mentioned in the manuscript, validation against in situ measurements is part of future work. The LWP obtained from the cloud radar 89 GHz passive channel can be assessed at other sites with a similar cloud radar and a dedicated microwave radiometer. Updated the manuscript on line 89: Standard Cloudnet instrumentation requires a dedicated multichannel microwave radiometer (MWR) on site, but in certain atmospheric conditions a single-channel MWR is able to provide LWP with sufficient accuracy. Figure 5 shows a comparison of the LWP from a similar RPG-FMCW-94 cloud radar (Moisseev, 2024a) and a multichannel RPG-HATPRO-G5 microwave radiometer (Moisseev, 2024b) in Hyytiälä, Finland, around 680 km south of Kenttärova, at the same time as the PaCE 2022 campaign. Measured LWP values over 0.1 kg m$^{-2}$ show a good correlation with a mean difference of 7 g m$^{-2}$, but the single channel overestimates smaller values with a mean difference of 21 g m$^{-2}$.

[Figure]

Figure 5: Comparison of liquid water path (LWP) hourly average from a single-channel microwave radiometer of a RPG-FMCW-94 cloud radar and a multichannel RPG-HATPRO-G5 microwave radiometer in Hyytiälä at the same time as the PaCE 2022 campaign. Data was only available for the end the campaign from 2 to 15 December.

2. **Comment:**

Also, an illustration of the VOODOO results would be helpful. The issue of missing liquid layers due to lidar attenuation is well known and VOODOO provides a valuable approach for the situations. Due to its still experimental stage, it would be good, to show and discuss the results of this method, for users inexperienced with VOODOO.

**Response:**

We agree that VOODOO should be discussed in more detail. We added a figure to illustrate the method and extended its description on line 156: This probability can then be used in the categorization of liquid pixels. Figure 6 shows a comparison of the Cloudnet classification using the standard method and the VOODOO method. The standard method detects less supercooled liquid than VOODOO and fails to identify any liquid above 3 km due to lidar attenuation. Under optically thick cloud conditions like this, VOODOO improves the standard classification, but more validation work is needed before it can be used operationally.

[Figure]

Figure 6: Example of the Cloudnet target classification on 24 September 2022 using **(a)** standard Cloudnet method and **(b)** VOODOO method.

**Specific comments**

1. **Comment:**

Line 67: Change the sentence "...up to a height of 15 km height" to "...up to a height of 15 km" to avoid repetition.

**Response:**

Corrected.

2. **Comment:**

Line 73: "attenuated backscatter cofficient" should be corrected to "attenuated backscatter coefficient.".

**Response:**

Corrected.

3. **Comment:**

Line 163: Add a comma before "and higher-level derived synergetic geophysical products.".

**Response:**

Corrected.

4. **Comment:**

Line 182: Change "according the FAIR principles" to "according to the FAIR principles.".

**Response:**

Corrected.

---

## Author Comment (AC2)

**Minor comments**

1. **Comment:**

   The importance of the site: The author mentioned the importance of an intensive field campaign over the Arctic and Subarctic region in the introduction, but failed to connect it to their campaign site: Kenttärova. I understand the site is situated in a Subarctic region, but the site description lacks the importance of the site from the cloud remote sensing perspective. The author needs to mention why the location is strategic to establish a cloud remote sensing campaign over there. It can be done by a brief discussion of the cloud climatology over the site and the necessity of studying those clouds and their interaction with aerosols using cloud remote sensing data. If there is already a detailed description in other studies please mention the reference.

   **Response:**

   We elaborated the site description on line 37: Cloud remote measurements conducted at Kenttärova complement the in situ observations conducted at the summit of the fell. The top of the fell is occasionally inside a cloud, offering possibilities to study the same cloud via both measurement principles. A more comprehensive description of Kenttärova, as well as the general weather and cloud conditions in the Pallas region, can be found in Hatakka et al. (2003) and Lohila et al. (2015).

2. **Comment:**

   Quality control of any data set is very crucial. The cloudnetpy-qc software that the author mentioned, is a Python package implementing the Cloudnet processing scheme based on the study of Illingworth 2007. The author mentioned Doppler wind lider performed a background noise measurement once per hour, which was used to correct the measurements during data post-processing. Apart from this information, nothing is provided in the present study or the Documentation on the Cloudnet site. For example; do the authors use wind lidar or LDR from radar to screen out the insects? Therefore, I suggest the authors to be more elaborative about the criteria and methodology they have used for the quality control of each instrument. The authors can also add it as documentation in the Cloudnet Data Portal and mention the reference in the present study.

   **Response:**

   Background measurements are only used to correct the Doppler lidar measurements, using the methodology described in Manninen et al. (2016) and Vakkari et al. (2019); insects are filtered using the radar LDR. Updated on line 104: Additionally, the instrument performed a background noise measurement once per hour, which were used to correct the Doppler lidar backscatter coefficient measurements.

   In the PaCE 2022 data set, the Doppler lidar products are complementary to the more mature products. In the future, the Doppler lidar measurements could be used in synergetic

Cloudnet products and to generate novel geophysical products. Updated on line 111: The Doppler lidar products were generated as complementary data for the PaCE 2022 campaign and were not used in the processing of the synergetic geophysical products. Although the Doppler lidar attenuated backscatter coefficient could be used in synergetic products, the two ceilometers offered higher temporal resolution due to their non-scanning operation.

3. **Comment:**

Further, the cloud classification methodology is adopted from Hogan and O'Connor (2006). Hogan and O'Connor were unable to distinguish supercooled drizzle from ice, but the present study shows the category of 'ice and droplets'. Also, the target classes shown in the present study are different than Hogan and O'Connor's (2006). A systematic explanation is necessary on how the authors are achieving those 11 classes. If the quality control and cloud classification are two general methods used in the Cloudnet site for the stations with Cloud Remote sensing facility, then adding the details as a document in the site would be useful for the readers of the present study as well as the Cloudnet users.

**Response:**

The methodology is the same as in Hogan and O'Connor (2004) with some revised methods in CloudnetPy (e.g. insect detection, Tukiainen et al., 2020). Clarified this on line 131: Then, each data point, or pixel, is categorized for the presence of liquid droplets, falling hydrometeors, freezing temperature, melting ice particles, aerosols, and insects (Hogan and O'Connor, 2004). A single pixel may contain several categories at once. Based on the categorization, a simpler target classification is derived where each pixel is assigned to one of 11 classes: "clear sky", "aerosols & insects", "insects", "aerosols", "melting & droplets", "melting ice", "ice & droplets", "ice", "drizzle & droplets", "drizzle or rain", or "droplets".

4. **Comment:**

All the derived products depend on the radar reflectivity. Hence, the radar reflectivity offset calibration is essential. Since the 94 GHz radar is highly attenuated to rain, and hail, and the authors have no option to check the offset due to the lack of any Disdrometer observations, they need to keep that in mind while validating with in-situ observations, and also the data need to be 'flagged' during heavy rain condition. Further, for vertically looking radar, mispointing could be an issue and it can lead to high bias to Doppler Velocity. I suggest the author discuss how they took care of the mis- pointing calibration of the radar.

**Response:**

We added a paragraph explaining potential sources of error after line 97: It should be noted that the cloud radar was not specifically calibrated for the PaCE 2022 campaign using established methods such as those described by Toledo et al. (2020) or Jorquera et al. (2023). Furthermore, no disdrometer was available on site at the time to monitor or verify the calibration. Nevertheless, since the instrument had been recently acquired and a liquid nitrogen calibration was performed a month before the campaign, it is reasonable to assume

that the radar reflectivity did not exhibit any significant bias during the observation period. One remaining potential source of error is radar mispointing. Since the cloud radar used in this study lacked a scanning unit, its vertical alignment relied solely on its internal tilt sensor. Although mispointing is generally less problematic for zenith-pointing cloud radars than for satellite-based or scanning weather radars, its impact should still be quantified. Identifying and correcting radar mispointing in the PaCE 2022 dataset is planned for future work.

5. **Comment:**

In regard to estimating droplet effective radius, Frisch et al. 2000 study is limited to stratus cloud only. Does the present study also focus on the stratus clouds while estimating droplet effective radius? If so, how the authors are classifying different types of clouds?

**Response:**

The reviewer is correct that the method described in Frisch et al. (2000) is only suitable for non-drizzling liquid clouds, and should not be applied to clouds containing drizzle (without modification of the original algorithm). The categorization algorithm used in Cloudnet identifies the presence or absence of drizzle in liquid layers, using the combination of lidar and radar profiles, and thus drizzle-free liquid layers can be identified satisfying the conditions necessary for applying the Frisch et al. (2000) algorithm.

6. **Comment:**

The goal of comparing cloud fractions with the ECMWF IFS model is not clear. The model having very low resolution clearly could not capture all the cloud features and hence nowhere it have any similarity with the observation in Figure 3. In spite, I would suggest the authors keep the estimated ice and liquid water content in the middle and the effective radius at the bottom panel in that figure for the same example.

**Response:**

Motivated comparison with the model on line 148: One of the aims of Cloudnet is the evaluation of NWP models on properties such as cloud fraction, and liquid and ice water contents (Illingworth et al., 2007). Figure 3 was not modified as suggested as it remains relevant to the discussion.

**Specific comments**

1. **Comment:**

Please mention if all the heights discussed in the present study are above mean sea level or ground level.

**Response:**

We agree this was a bit ambiguous. Changed:

(a) "up to 15 km height" to "up to a range of 15 km" on line 67.

(b) "up to an altitude of 12 km" to "up to a range of 12 km" on line 101.

(c) "height (km)" to "height (km AGL)" in y-axis labels of Fig. 3.

(d) "height" to "height above ground level" in Fig. 4 caption.

Other references to "altitude" or "height" should be clear.

2. **Comment:**
   In lines 84-85, I understand by 'large atmospheric targets' the authors meant compared to wind lider, but W-band radar would be attenuated for light rain, so please modify the line as 'making it sensitive to relatively large atmospheric targets such as ice particles, cloud droplets, and insects.'

   **Response:**
   Corrected.

3. **Comment:**
   In Figure 2. please mention in the caption, the time axis over which the parameters are averaged. Since the authors are showing data for 3 months, the radar reflectivity and also other parameters might be hourly or half-hourly averaged, so please mention that.

   **Response:**
   Updated Figure 2 to show the data in hourly averages and mentioned this in the caption.

4. **Comment:**
   Modify the Figure 4 to include the y-tick label 100 %. Does the exclusion of 'clear sky' cause the first column below 100%?

   **Response:**
   Figure 4 was modified to include 'clear sky' so that classes add up to 100 %.

[Figure]

Figure 4: Distribution of target classes at different heights above ground level during the PaCE 2022 campaign.

5. **Comment:**

What does it signify by 'Melting & droplets'? Does it quantify the ice that is melting near the bright band? There is no occurrence of 'Melting & droplets' in Figure 4, is it because the radar is not attenuation corrected for the melting layer and hence the use of the melting layer to detect melting ice and droplets is not possible in this study?

**Response:**

In the Cloudnet target categorization, more than one target type can be diagnosed for any individual pixel. The 'Droplets' target signifies that liquid water droplets have been diagnosed (usually by the lidar), and the 'Melting' target signifies that melting ice particles have been diagnosed (from radar LDR). It is possible for these two targets to co-exist in the same pixel, both in reality and in the categorization scheme. It should be noted that this combination is quite rare in our dataset (0.06 % of pixels)!

---

## Editor Decision (ED1)

**Editor's comment**

May 22, 2025

The authors present a valuable data set in a concise and clear way. Previous revisions have made improvements, but I do have some minor suggestions and comments to consider before publication.

**1 General comments**

**Line 25:** "Northern Finland" should be "northern Finland". It could be useful to add a sentence on what you mean with cold and clean or also cite some overview paper of the location like Lohila et al. 2015.

**Line 29:** Do you have references for these three PaCE campaigns? I would suggest to add them if you have them.

**Line 34:** "Northern Finland" should be "northern Finland".

**Line 71:** "up to about $7.6\,\mathrm{km}$" should be "up to a range of about $7.6\,\mathrm{km}$".

**Line 99:** "single channel" should be "single-channel". There is no good correlation, please change this to indicate the actual correlation coefficient and call it a positive correlation. It would sound a bit more clear, if you split the sentence into two: "Measured LWP values... $7\,\mathrm{g\,m^{-2}}$. For smaller values the single-channel passive radiometer measured a higher LWP with a mean difference of $21\,\mathrm{g\,m^{-2}}$."

**Line 135:** "multichannel" should be "multi-channel".

**Line 142:** You should add this as a citation and add a date to the link for when you last accessed it: `https://www.ecmwf.int/en/publications/ifs-documentation`, last access: XX Month YYYY. If applicable you should also add authors and year of publishing, as you did for the GitHub repositories.

**Line 147:** "2022 PaCE" should be "PaCE 2022".

**Line 168:** You adjusted the wording earlier in line 156 to categorization. I would suggest to do the same here.

**Line 180:** "machine learning-based" should be "machine-learning based".

**Line 328:** Toledo 2020 is cited as a discussion paper and should be upgrade to the final published version (`https://doi.org/10.5194/amt-13-6853-2020`).

**2 Comments regarding figures**

**Figure 2:** I am not sure, but it looks like the hatched areas are added outside of matplotlib. If you can, I would suggest to add hatched areas directly to the plot via Matplotlib. The following is a small snippet:

```
ax.axvspan(start_time, stop_time, color='none', hatch='/')
```

$y$-label of panel (b) does not contain the quantity (LWP).

**Figure 5:** "multichannel" should be "multi-channel"

**References**

Lohila, A. et al. (2015). "Preface to the special issue on integrated research of atmosphere, ecosystems and environment at Pallas". In: *Boreal Environment Research*. Vol. 20. 4, pp. 431–454.

---

## Author Response (AR2)

**General comments**

1. **Comment:**
   Line 25: "Northern Finland" should be "northern Finland". It could be useful to add a sentence on what you mean with cold and clean or also cite some overview paper of the location like Lohila et al. 2015.

   **Response:**
   We revised the paragraph and added another citation describing the Pallas environment.

2. **Comment:**
   Line 29: Do you have references for these three PaCE campaigns? I would suggest to add them if you have them.

   **Response:**
   This is the first paper on the remote sensing measurements during the PaCE campaigns. Publications on the in situ measurements during the earlier campaigns are cited in the manuscript (e.g., Kivekäs et al., 2009; Anttila et al., 2012; Doulgeris et al., 2023).

3. **Comment:**
   Line 34: "Northern Finland" should be "northern Finland".

   **Response:**
   Corrected.

4. **Comment:**
   Line 71: "up to about 7.6 km" should be "up to a range of about 7.6 km".

   **Response:**
   Corrected.

5. **Comment:**
   Line 99: "single channel" should be "single-channel". There is no good correlation, please change this to indicate the actual correlation coefficient and call it a positive correlation. It would sound a bit more clear, if you split the sentence into two: "Measured LWP values... 7 g m$^{-2}$. For smaller values the single-channel passive radiometer measured a higher LWP with a mean difference of 21 g m$^{-2}$."

   **Response:**
   Corrected the text as suggested and updated Fig. 5 (population correlation coefficient $\rho$ changed to sample correlation coefficient $r$).

6. **Comment:**
   Line 135: "multichannel" should be "multi-channel".

   **Response:**
   Corrected.

7. **Comment:**
   Line 142: You should add this as a citation and add a date to the link for when you last accessed it: https://www.ecmwf.int/en/publications/ifs-documentation, last access: XX Month YYYY. If applicable you should also add authors and year of publishing, as you did for the GitHub repositories.

   **Response:**
   Added proper citation with access information.

8. **Comment:**
   Line 147: "2022 PaCE" should be "PaCE 2022".

   **Response:**
   Corrected.

9. **Comment:**
   Line 168: You adjusted the wording earlier in line 156 to categorization. I would suggest to do the same here.

   **Response:**
   The categorization and target classification are different. Clarified the text to first discuss the categorization, then geophysical properties derived from the categorization, and finally the target classification also derived from the categorization.

10. **Comment:**
    Line 180: "machine learning-based" should be "machine-learning based".

    **Response:**
    Corrected.

11. **Comment:**
    Line 328: Toledo 2020 is cited as a discussion paper and should be upgrade to the final published version (https://doi.org/10.5194/amt-13-6853-2020).

    **Response:**
    Corrected.

**Comments regarding figures**

1. **Comment:**

   Figure 2: I am not sure, but it looks like the hatched areas are added outside of matplotlib. If you can, I would suggest to add hatched areas directly to the plot via Matplotlib. The following is a small snippet:

   ```
   ax.axvspan(start_time, stop_time, color='none' , hatch='/')
   ```

   y-label of panel (b) does not contain the quantity (LWP).

   **Response:**
   Corrected.

2. **Comment:**

   Figure 5: "multichannel" should be "multi-channel"

   **Response:**
   Corrected.